# Intercalated Poly (2-acrylamido-2-methyl-1-propanesulfonic Acid) into Sulfonated Poly (1,4-phenylene ether-ether-sulfone) Based Proton Exchange Membrane: Improved Ionic Conductivity

**DOI:** 10.3390/molecules26010161

**Published:** 2020-12-31

**Authors:** Murli Manohar, Prem P. Sharma, Dukjoon Kim

**Affiliations:** School of Chemical Engineering, Sungkyunkwan University, Suwon, Gyeonggi 440-746, Korea; madhavcsmcri87@gmail.com (M.M.); premsharma15@gmail.com (P.P.S.)

**Keywords:** polymer electrolyte membranes (PEMs), SPEES, proton conductivity, PMPS

## Abstract

A series of hybrid proton exchange membranes were synthesized via in situ polymerization of poly (2-acrylamido-2-methyl-1-propanesulfonic acid) PMPS with sulfonated poly (1,4-phenylene ether-ether-sulfone) (SPEES). The insertion of poly (2-acrylamido-2-methyl-1-propanesulfonic acid) PMPS, between the rigid skeleton of SPEES plays a reinforcing role to enhance the ionic conductivity. The synthesized polymer was chemically characterized by fourier-transform infrared spectroscopy (FT-IR) and nuclear magnetic resonance ^1^H NMR spectroscopy to demonstrate the successful grafting of PMPS with the pendent polymer chain of SPEES. A variety of physicochemical properties were also investigated such as ion exchange capacity (IEC), proton conductivity, water uptake and swelling ratio to characterize the suitability of the formed polymer for various electrochemical applications. SP-PMPS-03, having the highest concentration of all PMPS, shows excellent proton conductivity of 0.089 S cm^−1^ at 80 °C which is much higher than SPEES which is ~0.049 S cm^−1^. Optimum water uptake and swelling ratio with high conductivity is mainly attributed to a less ordered arrangement polymer chain with high density of the functional group to facilitate ionic transport. The residual weight was 93.35, 92.44 and 89.56%, for SP-PMPS-01, 02 and 03, respectively, in tests with Fenton’s reagent after 24 h. In support of all above properties a good chemical and thermal stability was also achieved by SP-PMPS-03, owing to the durability for electrochemical application.

## 1. Introduction

Due to the presence of an inbuilt property for converting chemical energy into electrical energy, fuel cell technology is gaining huge attention for portable as well as stationary objects [1,2]. Specifically, proton exchange membrane fuel cell (PEMFC) has been placed in the center because of adaptable size and high efficiency with low operational temperature [3,4,5]. On the other hand, environmental pollution caused by massive exploitation of fossil fuels and restrictions imposed by several organizations are the demanding factors for viable and green technology that is far better than the conventional methods of power production [6]. Nevertheless, being feasible, the deployment of PEMFC is still hindered because of the high cost of the ionomers, fuel crossover and long-term durability [7].

To address the above issue several polymer electrolyte membranes (PEMs) have been synthesized from a variety of sulfonated aromatic polymers such as poly (arylene ether sulfone ketone), polysulfone and polyimides for their application in proton exchange membrane fuel cell (PEMFC), because of their rigid backbone structures which are quite stable in thermal and mechanical aspects [8,9,10,11,12]. On the other hand, several researches have been finished in order to optimize the membrane properties which could attest themselves as best for fuel cell applications such as chemical and physical modifications viz grafting, cross-linking, pore filling and synthesis of composite membranes [13,14,15,16].

The sulfonated form of poly (1,4-phenylene ether-ether-sulfone) is known to be a promising aromatic polymer for various electrochemical applications, that has been broadly studied due to its outstanding thermal, mechanical and chemical stability. However, it also shows similar behavior to that of Nafion because of optimization regarding high proton conductivity that could be achieved by varying the degree of sulfonation. However, varying the sulfonation degree directly affects its dimensional stability due to other membrane properties deteriorating. To sort out these outcomes, the intercalation of PMPS onto the polymer backbone of sulfonated poly (1,4-phenylene ether-ether-sulfone) (SPEES) could be beneficial to enhancing both proton conductivity and flexibility of the membrane. In PMPS, the sulfonic acid groups attached within aliphatic chains are predictable to support the transportation of ions and induce lower crystallinity in the polymer backbone to reduce the brittleness by intercalation. Being an aliphatic and polar monomer, PMPS is a trend nowadays for the preparation of polyelectrolyte membranes for fuel cells and electrolysis [17,18,19].

Thus, the trade-off between high ionic conductivity with low water uptake and high mechanical stability can be maintained by tailor made hybrid membranes by grafting PMPS onto the SPEES structure. A high value of ionic conductivity was achieved by SP-PMPS hybrid membrane which is higher than that of SPEES. Experimental results prove the consistent enhancement of proton conductivity with temperature which is excellent for fuel cell application which can be operated at various temperatures.

## 2. Results and Discussions

### 2.1. Chemical and Structure Analysis

The chemical and molecular structure was analyzed to confirm the successful grafting of PMPS onto the polymer backbone of SPEES. FT-IR spectra of different SP-PMPS are depicted in Figure 1. The stretching vibration due to aromatic -C-C- present inside the benzene ring occurred at 1470–1590 cm^−1^ while at peak due to -OH groups attached to sulfonic group occur in a range from 3000 to 3140 cm^−1^. The characteristics peak corresponding to O=S=O occurred in frequency range 1110–1240 cm^−1^. We also observed the diminishment of the peak attributed to the hydroxyl group in cases of SP-PMPS-02 and 03, which could be due to the grafting of PMPS. On the other hand, the conversion of PEES into SPEES was also confirmed by FT-IR analysis by an intense peak at 1110 cm^−1^ as shown in Appendix A.

In addition to that, confirmation regarding the successful grafting of PMPS onto SPEES was also investigated by ^1^H-NMR spectroscopy (Figure 2). A chemical shift value around 7–8 ppm corresponds with the presence of aromatic protons where chemical shift value around 3.5 ppm and 2.5 ppm is associated with the protons associated with sulfonic acid group to both aromatic ring of benzene and aliphatic skeleton of PMPS, respectively. Above chemical shift values confirm the presence of PMPS and SPEES. Furthermore, ^1^H NMR spectra of SPEES is also depicted in Appendix A, representing the presence of aromatic protons associated with aromatic ring in the region 6–8 ppm.

### 2.2. Thermal Stability of SP-PMPS Membrane

Thermal stability for all the synthesized membranes was measured by thermogravimetric analysis (Figure 3). All prepared membranes displayed characteristic three-step weight loss. The stability of the membrane was quite excellent up to 100 °C as observed on the graph, but first weight loss occurred in the region of 150 °C due to the absorbed bound water content. It was more in SP-PMPS than SPEES because of the presence of more ionic clusters which attract moisture content due to the ionic interaction. Second weight loss was observed at ~250 °C which was due to the degradation of the functional groups. The first and second weight loss on a higher rate inside SP-PMPS than SPEES is strong evidence to justify the existence of more ionic domains which could be possible due to the successful grafting of aliphatic PMPS. The final weight loss observed at ~500 °C is due to the polymer backbone degradation. Thermal study of PMPS also shows a similar trend in which initial degradation starts from 80–100 °C due to absorbed moisture while high weight loss in the region 200–250 °C is due to the presence of sulfonic acid groups and backbone degradation starts at 325–500 °C [20,21]. Thus, this study reveals the durability of membranes over a high temperature range which could fit these membranes to various electrochemical applications which could be operated at high temperature.

### 2.3. Proton Conductivity

Proton conductivity is an important parameter for a polymer electrolyte membrane to be utilized for electrochemical application (Figure 4a). Herein, the existence of the sulfonic acid group will provide the hopping site for the proton for transportation. As it can also be controlled by maintaining the degree of sulfonation in the SPEES, achieving a higher sulfonation degree could weaken the polymer stability. Hence, synthesis of tailor-made membranes by using the combined effect of stability vs. conductivity has been shown by the present work. Adding PMPS onto the pendent polymer backbone of SPEES could cause the increment of ionic sites which facilitates the hopping mechanism for ionic transportation, but on the other hand, aliphatic side chains present in PMPS could provide integrity for stability. As can be observed in the mechanism in Figure 4b, the increment in the ionic cluster directly affords a positive impact on proton transportation thus we achieved a high conductivity without decreasing stability.

The SPEES membrane (Figure 5a), exhibits a smooth and tight surface while the SP-PMPS membrane (Figure 5b), exhibits an uneven surface, indicating that the phase-separated domains are interconnected with SPEES phases, which could be beneficial for ionic conductivity. The SPEES membrane SEM image shows the dense and homogenous nature without any cracks and holes. PMPS has higher hydrophilic nature because of the presence of sulfonic acid concentration more is higher than sulfonated SPEES. The sulfonic groups can generate larger electrostatic repulsive forces, thus the ion cluster size increased with increasing PMPS/SPEES molar ratio. Besides, the alkyl group of PMPS is hydrophobic, which can form hydrophobic micro-domains to decrease the hydrogen bonding interaction between hydrophilic polymeric chains and the polymer backbone.

The SAXS pattern of synthesized membrane was carried out to see the presence of ionic clusters and microphage separation in (Figure 6). As we can observe from the graph, no obvious microphage separation is present in SPEES but as we incorporated PMPS onto SPEES backbone the size of the ion cluster increases with distinct peak rise at q = 3.18 nm^−1^. These SP-PMPS hybrid membranes show more intense scattering peaks, suggesting larger spaces present in the ionic channels. The ionic cluster formation by PMPS would lead to an increase of the functional groups which enhance the proton conductivity, which would give a sudden impact on the performance during an electrochemical application. 

### 2.4. Water Uptake and Swelling Ratio

The water uptake phenomenon plays a key role for a membrane to be used in ionic transportation as it acts as a medium for the ions to be moved from one site to another through the membrane. The mechanical strength of a polymer electrolyte membrane is also dependent on water uptake behavior. Water uptake beyond a limit could cause deterioration in the freestanding property of the membrane. As we can see, the water uptake values gradually increase as we increase PMPS content, which could be associated with the presence of ionic clusters in the form of sulfonic acid bonds. The maximum value was achieved by SP-PMPS-03 which was 22.58, which is higher than SPEES which is 14.28 at 30 °C. The functional group present on the polymer backbone also provides the site for ion exchange which is represented as the ion exchange capacity. A gradual increase in the values of ion exchange capacity as we incorporated PMPS is a direct result of the increment in the sulfonic acid groups responsible for ion exchange sites present in the polymeric chain. As seen in Table 1, the value of ion exchange capacity is 1.70 meq g^−1^ for SPEES while it has increased to 2.71 meq g^−1^ as we increase PMPS content. Similar behavior was shown by the membranes when tested for swelling degree. High quantity of sulfonic acid group will cause penetration of water molecules through which the polymer chain expands. This triggered expansion of the polymer chain is responsible for the swelling behavior. However, the interionic interaction between the sulfonic acid molecules restricts the membrane from swelling up to a larger extent, thus a maximum swelling of 21.78% was achieved by SP-PMPS-03 at 30 °C. The corresponding value stating the physicochemical properties is shown in Table 1.

Additionally, a table of comparison related to the conductivity value of membranes from the current study with the reported literature is shown in Table 2.

### 2.5. Oxidative Stability

The durability of a membrane to be used for electrochemical application was measured by an oxidative stability test by immersing the samples of synthesized membranes into Fenton’s solution (3 ppm of FeSO_4_ into 3% H_2_O_2_, Figure 7).

The different membrane samples were tested for 72 h at 50 °C and changes occurring were checked by measuring the relative weight loss. The counterattack of formed peroxide radicals (^•^OH and ^•^OOH) is the main reason for the degradation of sulfonic acid groups as well as the polymer backbone. Synthesized membranes were quite stable up to 48 h and showing minimum weight loss. However, the degradation was higher afterwards as functional groups (-SO_3_H) associated with the polymer backbone must face extensive attack through formed radicals. Maximum weight loss occurred in SP-PMPS-03 as it exhibits a larger number of functional domains. The mechanism involved in the degradation of performance of polymer electrolyte membranes when counterattacked by radicals generated from Fenton’s reagent is also shown in Figure 8.

## 3. Materials and Methods

### 3.1. Materials

(PMPS and poly (1,4-phenylene ether-ether-sulfone) (PEES) were purchased from Sigma-Aldrich (St. Louis, MO, USA). Methanol, acetone, N, N-dimethylacetamide (DMAc), N, N-dimethylformamide (DMF), tetrahydrofuran (THF), N-methylpyrrolidone (NMP), dimethyl sulfoxide (DMSO) and sulfuric acid (H_2_SO_4_) were purchased from Daejung Reagents & Chemicals (Daejung, Korea).

### 3.2. Synthesis of SPEES (Sulfonated Poly Ether Ether Sulfone)

SPEES was synthesized by following the reported literature [25]. Briefly, poly (1,4-ether ether sulfone) was first kept in a vacuum oven at 70 °C for 12 h to eliminate moisture content. Then it was slowly dissolved in concentrated sulfuric acid (98%) and stirred at 45 °C for 9 h. The final and sulfonated product was precipitated in cold deionized (DI) water. Further, it was subjected to several washes with DI water to remove excess amounts of acid (H_2_SO_4_). The obtained neutral product was termed as sulfonated poly (1,4-ether ether sulfone) (SPEES). Further it was dried at 60 °C in an oven under vacuum. The yield of final product was ~90%.

### 3.3. Synthesis of PMPS-g-SPEES

Obtained product designated as SPEES (1.0 g) was dissolved in 10 mL DMAc solvent until the formation of a homogeneous and transparent mixture. Further, thionyl chloride (SOCl_2_) was added to the above mixture to obtain SPEES-Cl. The reaction was carried out for 5 h at 70 °C with continuous stirring. After completion, the product was precipitated in methanol and dried overnight at 60 °C. The obtained product was designated as SPEES-Cl.

In the second step SPEES-Cl was dissolved in DMAc to make it a 10 wt% solution. Simultaneously, PMPS was also dissolved in DMAc in separate vials. In the final step the PMPS solution was added dropwise into SPEES-Cl with different molar ratios viz 1:0.1, 1:0.2 and 1:0.3. The reaction mixture was stirred constantly for 12 h at 60 °C until the formation of sulfonamide bonds. Furthermore, on completion the reaction mixture was cast onto a clean glass plate and designated as SP-PMPS-01, SP-PMPS-02 and SP-PMPS-03, respectively, depending on the molar ratio of PMPS added to SPEES and as shown in reaction Scheme 1.

### 3.4. Chemical and Structural Analysis

The chemical assets, as well as structural analysis of grafted polymer, were characterized by FT-IR (Nicolet iS10, Thermo Fisher, 81 Wyman Street Waltham, MA 02451, USA) and ^1^H NMR (Unity Inova, Varian Palo Alto, CA, USA) analysis by using DMSO-d^6^ as solvent. The microstructure morphology of formed membrane was evaluated by Field Emission Scanning Electron Microscope (FE-SEM) (EM, Phillip XL30 ESEM-FCG, North Billerica, MA, USA). Stepwise weight loss of synthesized membranes was characterized by thermogravimetric analysis (TGA, Seiko Exstar 6000, Tokyo, Japan).

### 3.5. Physicochemical Properties of SP-PMPS

Water uptake of synthesized samples of membrane was measured by measuring the difference in weight (g) between the wet and dry states of the membrane. Briefly, samples having dimensions (3 × 1 cm) were placed into oven for 24 h at 100 °C and then dipped into DI water. Further, it was calculated by following Equation (1),
(1)$$WU\left(\%\right)=\frac{w_{1}-w_{o}}{w_{o}}\times 100\;\;\;\;\;\;\;\;$$
where *WU* (%) is the water uptake of membrane samples and $w_{o}$ and $w_{1}\;$are the weights of the dry and wet samples.

Proton conductivity of SM-PMPS membrane samples was calculated by using an impedance unit (Zive SP2 electrochemical workstation, Wonatech, Seoul, Korean) assembled with a 4-probe cell (BEKKTECH, Colorado, USA) within the frequency ranging from 1 Hz to 1 MHz at 5 mV at 100% relative humidity (RH) [24]. The proton conductivity was calculated from the following Equation (2),
(2)$$\sigma =\frac{L}{R\times W\times T}\;\;\;\;\;\;$$
where, σ is the proton conductivity $\left(S\;cm^{-1}\right)$, *L* is the distance between the reference electrodes (cm), *R* is the resistance of the ionomer (Ohm) and *W* and *T* are the width (cm) and thickness (cm) of samples, respectively.

### 3.6. Chemical and Oxidative Stability of SP-PMPS

Chemical and oxidative stability of membranes was investigated by measuring both the residual weight percentage of the SP-PMPS in Fenton’s solution after treatment. The dry form of membranes were immersed in the Fenton’s solution (3 wt% H_2_O_2_, 4 ppm Fe^2+^) at 50 °C for 24 h. Afterwards, the samples were washed several times with DI water and then dried at 80 °C under vacuum. The residual weight percentage (RW) was calculated by the difference in the weight of the sample before (*m_b_*) and after treatment (*m_a_*) from Equation (3).
(3)$$RW\left(\%\right)=\frac{m_{a}}{m_{b}}\times 100\;\;\;\;\;$$

### 3.7. SAXS Analysis of SP-PMPS Membrane

SAXS (Small Angle X-ray Scattering) analysis was carried out to confirm the existence of microphase separation in SPEES and SP-PMPS membranes. The occurrence of intense peaks in the SAXS graph is mainly attributed to the presence of ionic cluster dimensions which is responsible for migration of protons. SAXS analysis pattern of pristine SPEES and SP-PMPS membranes in which ion cluster dimensions are measured by the given Equation (4),
*d* = 2π/*q*(4)
where *q* is the scattering vector, which is equal to 4π/λsin θ where 2θ belongs to the scattering angle and λ is the wavelength of the X-ray passed. 

## 4. Conclusions

An elevation in the proton conductivity was successfully attained by grafting aliphatic polymer onto SPEES by sulfonamide bond formation. SP-PMPS-03 exhibits a proton conductivity value of 0.089 S cm^−1^ at 80 °C. The chemical and molecular structure of synthesized hybrid membrane was characterized by ^1^HNMR and FT-IR spectroscopical techniques. Membranes show optimum water uptake of 22.58% with IEC value 2.71 meq g^−1^ which is quite good to be employed in electrochemical applications. Synthesized membranes are stable at high temperature range of ~100 °C as shown through TGA thermogram. Oxidative stability tests conducted at 50 °C showed quite a stable nature with minimum weight loss up to 24 h which is 93.35, 92.44 and 89.56%, for SP-PMPS-01, 02 and 03, respectively, whereas as it is 95.25% for pristine SPEES, indicating the decent increment of more vulnerable sites as we increase PMPS content in the form of functional groups. Optimal water uptake with high conductivity and high thermal and oxidative stability for successfully synthesized SP-PMPS hybrid membrane is proving the beneficial effect of an aliphatic polymer on the performance of SPEES. All above discussed properties suggest the promising behavior of synthesized membranes for suitable electrochemical applications.

## Data Availability

Data availability will be provided from authors.

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
