# Peer review of "Intercalated Poly (2-acrylamido-2-methyl-1-propanesulfonic Acid) into Sulfonated Poly (1,4-phenylene ether-ether-sulfone) Based Proton Exchange Membrane: Improved Ionic Conductivity"

_molecules, 2020, doi:10.3390/molecules26010161_

Round 1

Reviewer 1 Report

This paper presents and interesting alternative for PEM based on the intercalation of PMPS with SPEES in order to get a suitable polymeric membrane that combines high proton conductivity and not too excess swelling when hydrated, which could attempt against the mechanical properties when applied into a real fuel cells (temperature, electrode environment, humidity...). The paper is suitable for publication after addressing some minor comments:

  1. Which is the relative humidity used in the pronton conductivity measurement. Let's recall that this conditions strongly impacts on the conductivity for sulfonated systems based on the mechanism for proton conduction. At the same time, a reference to this mechanism would be nice.
  2. SEM images show some irregularities in the membrane surface, any comment or discussion on an eventual impact during the formation of a membrane-electrode-assembly? Might this hinder the assembly process?
  3. Oxidative stability. First, I consider that a comment on this should be included in the abstract and conclusions. The results display that the membrane are unstable and this should be undoubtedly improved for application in fuel cells, especially thinking on long-term applications. A comment on this would be nice.
  4. For future works, consider to measure the mechanical properties.

After discussing these aspects, the paper could be suitable for publication.

Author Response

Reviewer 1: Comments and Suggestions for Authors

This paper presents and interesting alternative for PEM based on the intercalation of PMPS with SPEES to get a suitable polymeric membrane that combines high proton conductivity and not too excess swelling when hydrated, which could attempt against the mechanical properties when applied into a real fuel cells (temperature, electrode environment, humidity...). The paper is suitable for publication after addressing some minor comments:

  1. Which is the relative humidity used in the proton conductivity measurement. Let us recall that this conditions strongly impacts on the conductivity for sulfonated systems based on the mechanism for proton conduction. At the same time, a reference to this mechanism would be nice.

Answer: Thank you for comment on the relative humidity condition during proton conductivity measurement. The conductivity for all the membranes were measured with 100 % (RH) relative humidity. To support the experimental data a reference has been added to the manuscript (page no-5 and line no- 109-110).

    2. SEM images show some irregularities in the membrane surface, any comment or discussion on an eventual impact during the formation of a membrane-electrode-assembly? Might this hinder the assembly process?

Answer: Thank you for this informative and valuable comment. The phase segregated behavior and formation of hydrophilic domain on a larger side is due to incorporation of PMPS and it is also supported by SAXS analysis in the revised manuscript on a smaller scale. Further, we consider proper conditioning of the membrane before using it for membrane electrode assembly to avoid improper and ununiform contact of the catalyst over the surface of the membrane.

    3. Oxidative stability. First, I consider that a comment on this should be included in the abstract and conclusions. The results display that the membrane is unstable, and this should be undoubtedly improved for application in fuel cells, especially thinking on long-term applications. A comment on this would be nice.

Answer: Thank you for the recommendation as guided in the comment, a sentence addressing oxidative stability values for the synthesized membrane has been added to abstract as well as in the conclusion section. Further, vigorous experiments are going on to optimize time of reaction and suitable condition to improve the oxidative stability for long term process that could be impactful for its operation in fuel cell with long term stability. (page no-1 line 20-22 and page no- 12 line 259-261)

    4. For future works, consider measuring the mechanical properties.

Answer: Thank you for this valuable suggestion and it will be added regarding future perspective.

Reviewer 2 Report

The authors have characterized a class of SP-PMPS based membranes.  It is an interesting way to improve the water uptake and ionic conductivity of the membrane. However, the following points need to be addressed before the acceptance of the paper for publication in Molecules.

  1. In experimental section of synthesis of PMPS-g-SPEES, do we keep the amount of SPEES-Cl the same and add different amount of PMPS? It is very confusing to write the molar ratios by 1:01. 1:02 and 1:03.
  2. The author should mark the characteristic FTIR peaks in Fig.1 for SP-PMPS-01, 02, and 03 for the audience to better illustrate the structural information. Also, the author should add FTIR spectrum of PEES and SPEES into Fig.1 for better comparison.
  3. It is better to add TGA information for PMPS in Fig.3.
  4. How does the conductivity of SP-PMPS membranes compare with the up-to-date values in literature. The author should provide the discussion for that.
  5. It is very speculative to argue that phase-separated domains are interconnected with SPEES phases. The SEM image is in micrometer scale while as the phase-segregated domain often lies in nanometer scale. The author should do SAXS experiments to verify their hypothesis.
  6. What is the acceptable water uptake in the literature for PEMFC application? The author should discuss the water uptake and IEC values with the literature.
  7. What is the degradation mechanism of SP-PMPS membrane when immersed into Fenton’s solution?

Author Response

Reviewer 2: Comments and Suggestions for Authors

The authors have characterized a class of SP-PMPS based membranes.  It is an interesting way to improve the water uptake and ionic conductivity of the membrane. However, the following points need to be addressed before the acceptance of the paper for publication in Molecules.

  1. In experimental section of synthesis of PMPS-g-SPEES, do we keep the amount of SPEES-Cl the same and add different amount of PMPS? It is very confusing to write the molar ratios by 1:01. 1:02 and 1:03.

Answer: Thank you for the suggestion and the molar ratios for different membranes have been changed from 1:01, 1:02 and 1:03 to 1:0.1, 1:0.2 and 1:03 for SP and PMPS, respectively. (page no-3, line no-86).

    2. The author should mark the characteristic FTIR peaks in Fig.1 for SP-PMPS-01, 02, and 03 for the audience to better illustrate the structural information. Also, the author should add FTIR spectrum of PEES and SPEES into Fig.1 for better comparison.

Answer: The characteristics peak regarding chemical structure in FTIR studies have been marked properly in the revised manuscript. (page no- 6, Figure no-1)

    3. It is better to add TGA information for PMPS in Fig.3.

Answer: According to the current suggestion it has been added to the TGA study section. (page no-7, line no-152-155)

    4. How does the conductivity of SP-PMPS membranes compare with the up-to-date values in literature? The author should provide the discussion for that.

Answer: Thank you for this valuable suggestion. Well, a table including the comparison of conductivity values with state-of-the -art has been added to the revised manuscript. (page no- 11, Table no-2).

    5. It is very speculative to argue that phase-separated domains are interconnected with SPEES phases. The SEM image is in micrometer scale while as the phase-segregated domain often lies in nanometer scale. The author should do SAXS experiments to verify their hypothesis.

Answer: Thank you for this comment to support the formation of phase segregated domain SAXS study has been carried out to support the current experimental study. (page no-9, line no- 187-193 and Figure no-6).

    6. What is the acceptable water uptake in the literature for PEMFC application? The author should discuss the water uptake and IEC values with the literature.

Answer: The acceptable value for water uptake for the membrane to be applicable for fuel cell application according to the literature has been listed in Table 2. The corelated study for water uptake and ion exchange capacity has been added in the experimental section. (page no- 10-11, line no-203-207).

    7. What is the degradation mechanism of SP-PMPS membrane when immersed into Fenton’s solution?

Answer: The mechanism showing degradation of SP-PMPS membrane with Fenton’s solution has been added to the revised manuscript. (page-12, line no-234-236 and Figure 8)

Round 2

Reviewer 2 Report

The paper can be published in its present form.